# Hypotaurine promotes glioma cell invasion by hypermethylating the Wnt5a promoter

**Hong Tian** ⓘ *, **Xiaoli Chu, Qi Guan, Juan Liu, Ying Liu**

Department of Oncology, The 4th People's Hospital of Shenyang, China Medical University, Shenyang, P. R. China

* sysytianhong@163.com

## Abstract

Glioma is a particularly lethal central nervous system tumor. Identifying the boundary between gliomas and normal tissues is difficult due to their infiltrative and invasive growth characteristics. This can result in the inevitable recurrence of the tumor after surgery. Preventing the residual tumor from growing or spreading is a major obstacle in treating gliomas. An earlier study suggested that hypotaurine could enhance the invasion of glioma cells while inhibiting the activity of demethylases. The hypotaurine synthesis-deficient U251 cell line usage showed a decrease in the cells' invasion capability. Analysis of gene expression profiles showed that reducing the activity of a critical enzyme in hypotaurine production, 2-aminoethanethiol dioxygenase (ADO), had a notable effect on the extracellular matrix-receptor interaction. Decreased intracellular ADO expression led to a significant increase in Wnt5a expression. Cells exposed to hypotaurine exhibited decreased levels of both intracellular Wnt5a protein and its corresponding mRNA. The observed characteristic was linked to increased methylation of the Wnt5a gene promoter, possibly due to hypotaurine's ability to inhibit demethylase enzymes. To sum up, the research showed that U251 cells lacking hypotaurine synthesis were susceptible to epigenetic changes, and Wnt5a seemed to function as a cancer inhibitor in this scenario. It would be beneficial to reevaluate this tumor suppressive effect in real tumor samples, which may contribute to the development of new glioma interference strategies.

## Introduction

Despite being less common than other types of brain tumors, glioma is the deadliest worldwide according to Wesseling and Capper [1]. Removing all of the diseased tissue surgically continues to be a major obstacle thus far. The presence of residual tumor tissue renders post-operative recurrence a near certainty, even in the context of extensive adjuvant chemotherapy. After a relapse, the outlook usually deteriorates, with more than 95% of patients experiencing a recurrence passing away within three years

**Data availability statement:** All relevant data are within the paper and its Supporting Information files.

**Funding:** Funding for this project was provided by the Natural Science Foundation of Liaoning Province (2020-MS-327).

**Competing interests:** The authors have declared that no competing interests exist.

[2]. Glioma invasion is a critical factor contributing to the poor prognosis of glioma patients. Gliomas have been categorized into four grades (I-IV) by the World Health Organization (WHO) according to how invasive they are. The higher the tumor grade, the more invasive the tumor [3]. Understanding the molecular mechanisms underlying the invasive behavior is essential for developing effective therapeutic strategies.

Metastasis of gliomas is a rare occurrence. Gliomas are characterized by their local infiltrative and invasive growth patterns. A multitude of mechanisms have been investigated in order to elucidate the invasive capabilities of gliomas. Most research has concentrated on the atypical stimulation or suppression of proteins within distinct signaling pathways, such as the mammalian target of rapamycin complex [4], the Wingless/Int1 (Wnt) proteins [5], and transforming growth factor-β [6]. These pathways have been shown to be closely related to metastasis and invasion in a variety of malignant tumors. Genetic factors were identified as the main cause of the dysregulated activation in most relevant pathways.

Apart from protein molecules, there has been significant focus on the roles of small molecular metabolites in controlling the invasion of glioma. IDH1 and IDH2 mutations are common in most lower-grade gliomas (grades II and III) and some grade IV gliomas, such as glioblastoma (GBM), according to Cohen et al. [3]. Under typical conditions, IDH facilitates the transformation of isocitrate into α-ketoglutarate (α-KG). Gliomas experience a significant increase in α-hydroxyglutarate (α-HG) production due to the loss-of-function mutation of IDHs, which leads to the conversion of isocitrate into α-ketoglutarate (α-KG) [7]. α-KG is a natural substrate for numerous α-KG-dependent dioxygenases. Certain enzymes that are part of the process of altering the genetic material through epigenetic mechanisms are 5-methylcytosine hydroxylases and histone lysine demethylases [8]. Because α-HG and α-KG have similar structures, α-HG can compete with α-KG for binding to the dioxygenases, leading to a decrease in enzymatic activities. The act of suppressing usually leads to a hypermethylation pattern in the genetic material and plays a role in the invasion of glioma to different extents, as shown by Flanagan et al. [9] and Jalbert et al. [10].

Hypotaurine is a precursor in the taurine biosynthesis pathway and has been implicated in various physiological and pathological processes, including oxidative stress and cancer progression. Recent studies suggest that hypotaurine may influence tumor invasion and metastasis, although its precise role remains unclear. According to a recent research, glioma tissues have shown increased levels of hypotaurine, which is an amino acid containing sulfur [11]. The intracellular content of the amino acid was found to correlate with tumor grades. Hypotaurine is important because it can block the binding of α-KG to various dioxygenases, such as 5-methylcytosine (5mC) hydroxylase, histone demethylase (H3K9), and histone demethylase (H3K4), in a way similar to α-HG [11,12]. Glioma cells lacking the ability to synthesize hypotaurine showed reduced invasion capability, but this could be reversed by supplementing with external hypotaurine [11].

Wnt5a belongs to the glycoprotein family of the Wnt pathway. It plays a role in various pathways related to development and cancer formation [13]. The specific function of Wnt5a in the development of tumors is still not fully understood but it exhibits

dual roles in cancer biology, functioning as a tumor suppressor in certain contexts and as a tumor promoter in others. The role of Wnt5a in glioma development needs to be further investigated.

This study investigates the role of hypotaurine, a sulfur-containing amino acid derivative, in glioma cell invasion, with a specific focus on its impact on Wnt5a gene expression and promoter methylation. By elucidating these mechanisms, we aim to provide novel insights into glioma progression and identify potential therapeutic targets. To clarify how hypotaurine may impact invasion, a glioma cell line lacking the ability to synthesize hypotaurine was used in the research. Analysis of gene expression profiles showed that hypotaurine suppressed the expression of Wnt5a. Additionally, it was discovered that increased levels of hypotaurine inside cells can boost the methylation of the Wnt5a promoter. Consequently, we postulate that hypotaurine exerts its inhibitory effect on demethylase activities, resulting in Wnt5a promoter hypermethylation and enhanced invasion ability of glioma cells.

## Materials and methods

### Reagents

Cysteamine, puromycin, hypotaurine, and 5-aza-2'-deoxycytidine were obtained from Sigma-Aldrich (Shanghai, China). Dimethyl sulfoxide (DMSO) was supplied by Solarbio (Beijing, China). 5-aza-2'-deoxycytidine was dissolved in DMSO and kept in liquid nitrogen.

### Cell and cell culture

Cells with impaired hypotaurine production, U251 cells with suppressed 2-aminoethanethiol dioxygenase (ADO) gene expression (ΔADO), were obtained from 3DMed Shanghai, China after being created using RNA interference. DMEM high glucose culture media containing 10% FBS and 0. 5 μg/ml puromycin were used. The cells were incubated at a temperature of 37 degrees Celsius in the presence of 5% carbon dioxide. Cells treated with 5-aza-2'-deoxycytidine at a concentration of 5 μM could survive for a period of 3 days. Every day, the new 5-aza-2'deoxycytidine culture was refreshed.

### Quantitative real-time PCR

Total RNA was extracted utilizing RNAiso Plus from TaKaRa, Dalian, China. First-strand cDNA synthesis was performed with the PrimeScript RT reagent kit and gDNA Eraser (TaKaRa). Quantitative real-time PCR was carried out on an Agilent Mx3000P system (Santa Clara, CA) employing the SYBR Premix Ex Taq kit (TaKaRa). The primers for the ADO gene were 5'-GGAGCACTGTTTCTCCCTTTT-3' and 5'-CAATCAAGAGGGCTTAGACGA-3'. For the Wnt5a gene, the primers used were 5'-GTGCAATGTCTTCCAAGTTCTTC-3' and 5'-GGCACAGTTTCTTCTGTCCTTG-3'. β-actin expression served as a normalization control, with primers 5'-CGAGGCCCAGAGCAAGAGAG-3' and 5'-CTCGTAGATGGGCACAGTGTG-3'. The relative expression levels were determined using the $2^{(-\Delta\Delta Ct)}$ method.

### Gene expression profile analysis

Novogene (Beijing, China) conducted RNA sequencing gene expression analysis using RNAseq. Differences in gene expression were identified between ΔADO cells and their control Vct cells. Three separate biological replicates of every cell line were utilized for gene expression analysis. The differentially expressed genes in the two groups were screened out, and the threshold was set as follows: $|log2(FoldChange)| > 1$ and qvalue < 0. 005. Subsequent analyses included comparing gene clusters, enriching GO terms, conducting KEGG enrichment analysis, and examining differential gene-protein interactions.

### Methylation-specific PCR (MSP)

The MSP response was conducted according to the method outlined in a previous study [14]. Genomic DNA was briefly isolated with DNAiso from TaKaRa. A portion of the DNA that was separated was exposed to sodium bisulfite. The

amplification process was carried out with the TaKaRa Ex Taq kit as per the provided guidelines. The primers used for detecting methylation sites were 5'-GTATTTTTCGGAGAAAAAGTTATGC-3' and 5'-AACCGCGAATTA'ATATA'AACGTC-3'. The primers for the sites without methylation were 5'-GGTATTT TTTGGAGAAAAAGTTATGTG-3' and 5'-'CAACCACAAATTAATATAAACATC-3'. The DNA fragments were separated using 1% agarose gel electrophoresis.

## Western blot analysis

Protein extraction was performed by isolating total protein with RIPA Lysis and Extraction buffer obtained from Thermo Fisher Scientific in Waltham, MA. Mini-PROTEAN TGX SDS-PAGE gels from Bio-Rad (Hercules, CA) were utilized for conducting protein electrophoresis. Abcam (Cambridge, UK) provided the antibodies against human Wnt5a, β-actin, and mouse, including a mouse anti-human antibody and a goat anti-mouse antibody.

## Hypotaurine quantification

Intracellular hypotaurine quantification was performed following the protocol outlined in a previous study [11]. Cells were harvested once they reached 70–80% confluence. Initially, the cells were rinsed with cold PBS three times before undergoing metabolite extraction with 1ml of a 21:79 mixture of methanol and water, followed by 2ml of chloroform. Following centrifugation, the liquid above the sediment in each specimen was transferred to a fresh tube and freeze-dried at a temperature of -50°C. Hypotaurine derivatization was performed using the AccQTag Ultra Derivatization Kit by Waters in Milford, MA, followed by quantitation with the 1290 Infinity ultra performance liquid chromatography system coupled to the 6460 Triple Quad mass spectrometry system by Agilent. Biological triplicates were used to analyze each cell line, with ion intensity used for relative quantitation after calibration against the dry weight of individual proteins.

## Results

### Reducing ADO gene expression resulted in a decrease in the amount of intracellular hypotaurine

RT-PCR analysis was conducted to confirm the successful knockdown of the ADO gene by verifying its expression. ΔADO U251 cells were discovered to have reduced expression of the ADO gene, as shown in Fig 1A). The ΔADOU251 also showed a decrease in intracellular hypotaurine levels (Fig 1B).

### ADO gene knockdown increased the expression of WNT5A

Analysis of RNA sequencing expression in the ΔADO and its empty vector cell line (Vct) revealed that 96 genes were increased and 218 genes were decreased due to the ADO gene suppression (Fig 2A). Analysis of pathway enrichment using gene expression data revealed that the most disrupted cellular process was the interaction between the extracellular matrix (ECM) and receptors. 2C). Wnt5a was one of the genes that showed the most differential expression in the ECM-receptor interaction process, as depicted in Fig 2D, p = 0. 0001).

### *The invasion capabilities of U251 cells were reduced by hypotaurine through the downregulation of WNT5A expression*

Cysteamine can be converted into hypotaurine via the ADO pathway, as demonstrated by Gao et al. in 2016 [11]. ΔADO and Vct U251 cells were cultured with or without 400 μM cysteamine. Incubation of Vct cells with cysteamine resulted in a significant increase in invasion ability (p = 0. 037), while the invasion ability of ΔADO cells remained unaffected by additional cysteamine incubation (p = 0. 817) (Fig 3A). Stimulation of the two cell lines with cysteamine resulted in a significant increase in intercellular hypotaurine specifically in Vct cells (p = 0. 031) (Fig 3B). The Vct cells showed elevated levels of hypotaurine within the cells, which was associated with reduced levels of Wnt5a protein and impaired expression of the

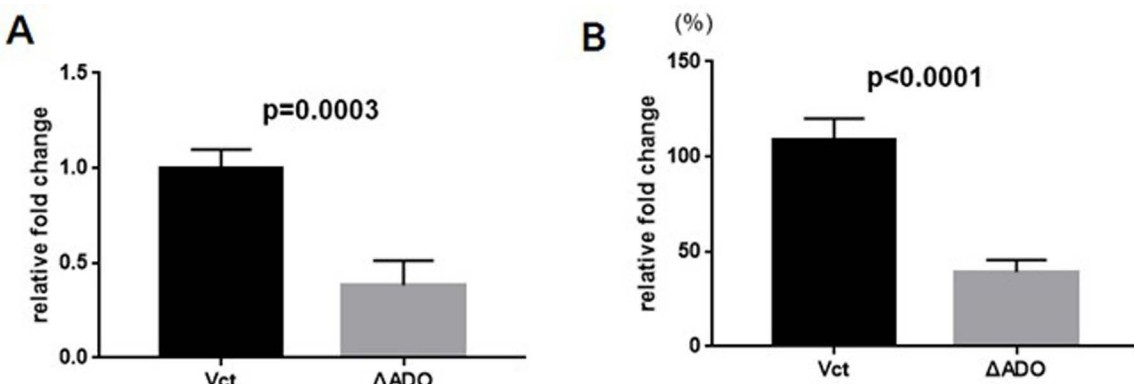

**Fig 1. Decreased hypotaurine levels within cells lacking the ADO gene in U251 cells.** RT-PCR was used to analyze the relative expression of the ADO gene. The ADO data was standardized in comparison to the Vct data. Comparing the hypotaurine content between the two cells at an intercellular level. The ADO data was standardized in comparison to the Vct data.

Wnt5a gene (Fig 3C and 3D). The distinct variations in Wnt5a gene and protein expression were not observed in ΔADO cells stimulated with cysteamine (Fig 3C and 3D).

### Hypotaurine increased methylation status of Wnt5a promoter and cell invasion & proliferation abilities

U251 cells were exposed to demethylation reagents 5-aza-2'-deoxycytidine for a period of 3 days. The methylation status of the Wnt5a promoter region was assessed with methylation-specific PCR, following the method outlined by Jiang et al. in 2017 [14]. The cells treated with 5-aza-2'-deoxycytidine exhibited a lower level of methylation in the Wnt5a promoter region compared to the cells treated with DMSO and those left untreated. After hypotaurine treatment, the ΔADO cells exhibited increased methylation of the Wnt5a promoter compared to untreated cells (Fig 4A). This demonstrated that hypotaurine has the ability to improve the methylation level of the Wnt5a promoter. Meanwhile, the increase of methylation at the promoter site of Wnt5a with hypotaurine treating which led to the decrease of Wnt5a expression promoted cell invasion and proliferation ability of U251 cells (Fig 4B and 4C). While the 5-aza-2'-deoxycytidine treatment showed the opposite results, the methylation degree of the promoter site of Wnt5a was reduced and the invasion and proliferation of U251 cells were inhibited.

## Discussion

Gliomas, especially those of advanced grades, are known for their invasive and infiltrative expansion. Unlike most solid tumors, glioma cells do not spread via blood vessels or lymphatic pathways [15]. Gliomas invade by cells migrating through the extracellular area of surrounding healthy brain tissue. In order to finish this procedure, glioma cells need to either break down or stick to the ECM [16,17]. The ADO gene has been demonstrated to decrease hypotaurine synthesis and compromise the invasion ability of U251 cells [11]. The RNA sequencing analysis revealed a notable impact of hypotaurine deficiency on the ECM-receptor pathway (Fig 2C). This suggested a strong correlation between hypotaurine and the invasive potential of glioma cells.

Wnt5a exhibits dual roles in cancer biology, functioning as a tumor suppressor in certain contexts and as a tumor promoter in others. For instance, in a study of neuroblastoma, Blanc et al. (2005) observed that metastatic neuroblasts exhibited downregulated Wnt5a expression compared to primary tumors [18]. In 2008, a research was carried out to examine healthy colon cells, colorectal cancer cells, and tumor cells from patients [19]. The results indicated that Wnt5a was commonly deactivated in cancerous colon cells. The observed reduction in Wnt5a expression was attributed to promoter

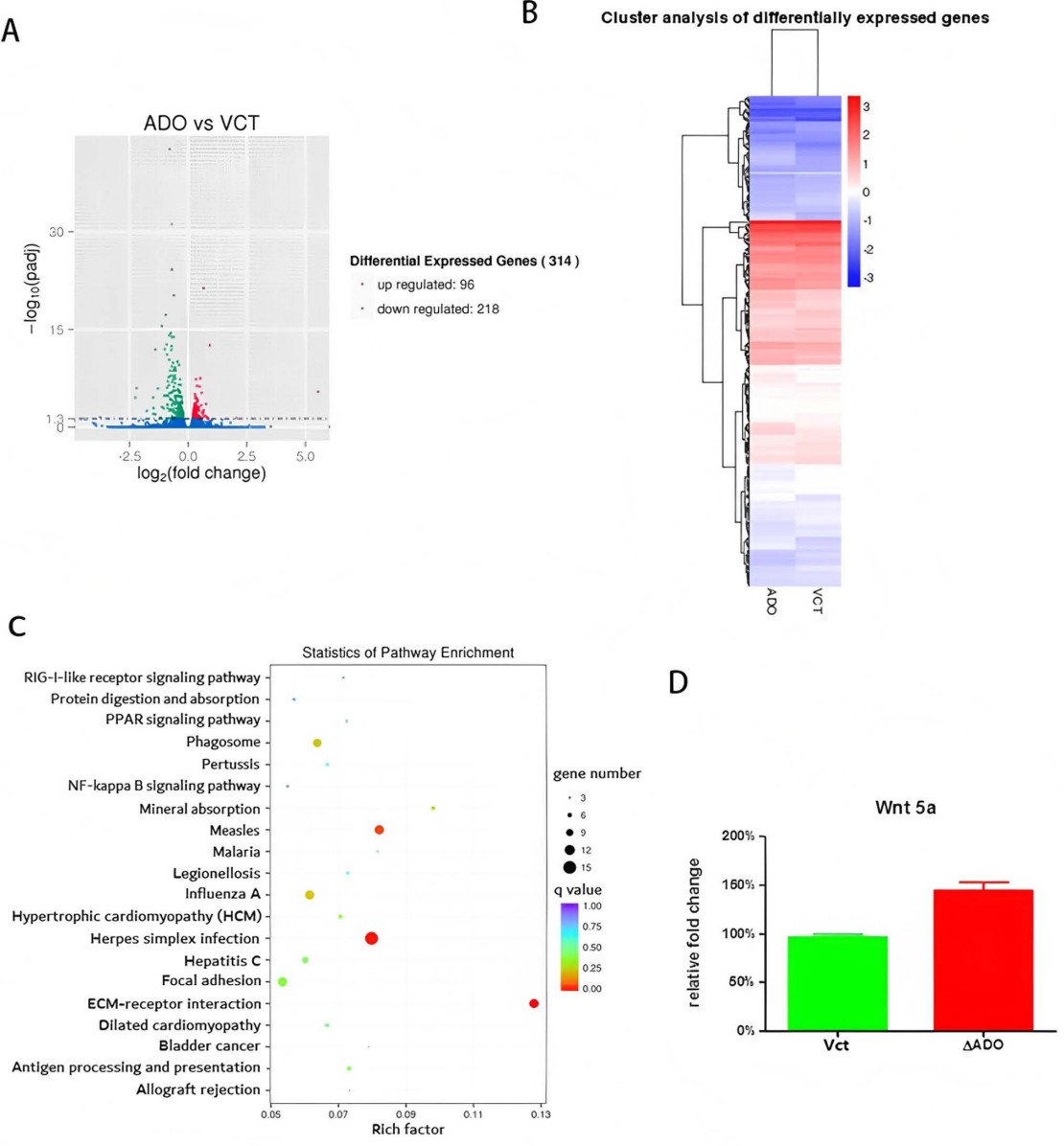

**Fig 2. Genes that are expressed differently between ΔADO and Vct cells.** The graph displaying genes with varying levels of expression. **(B)** Clustering analysis of differential genes. Gene expression enrichment analysis identified the pathways most significantly affected by the knockdown of the ADO gene. The sizes of the dots indicated the genes that were affected in the respective pathways. Wnt5a expression was markedly elevated in the genetically altered U251 cells lacking ADO compared to the control Vct cells (p = 0. 0001).

hypermethylation [19]. These findings indicated that Wnt5a may act as a tumor suppressor. On the other hand, the proof also suggested that Wnt5a has the potential to function as a promoter of tumors. In a research on non-small-cell lung cancer, it was shown that the growth, movement, and invasion capabilities of cancerous cells were hindered when Wnt5a was suppressed by microRNA [20]. A leukemia cell study also showed comparable findings, indicating that Wnt5a promoted

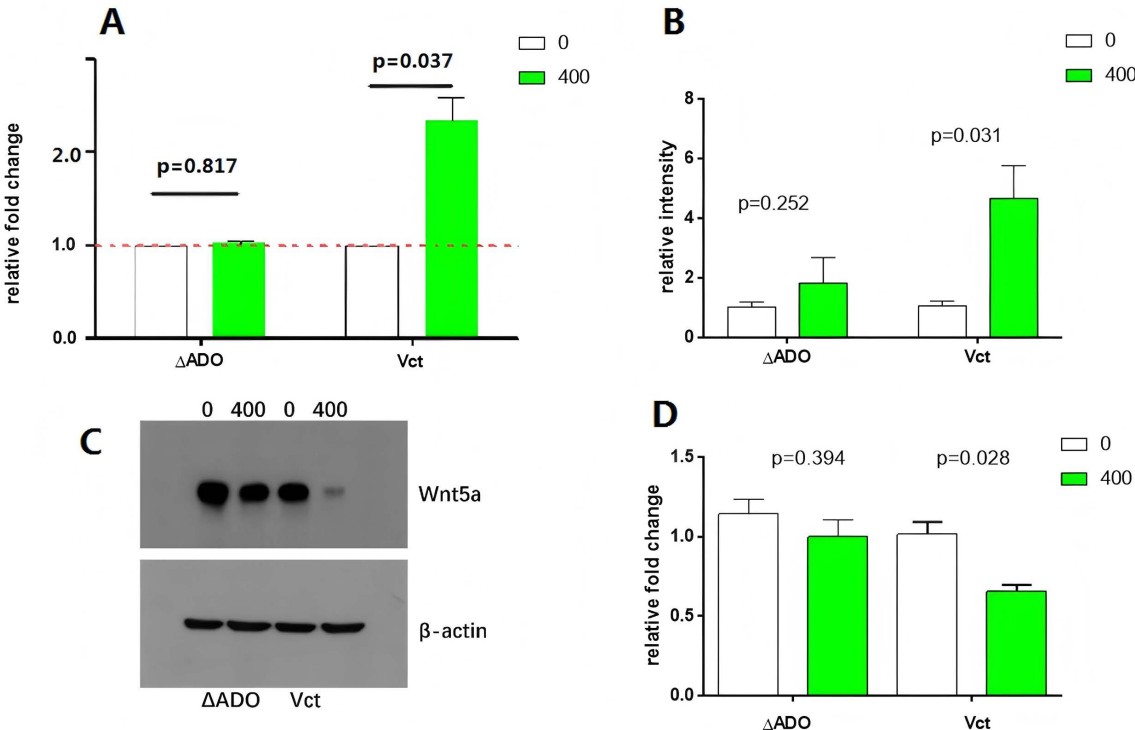

**Fig 3. The presence of hypotaurine during incubation may influence the expression of Wnt5a and the invasive capabilities of U251 cells.** ADO and Vct cells were separately cultured with or without 400 μM cysteamine. Cysteamine solely enhanced the invasive capacity of Vct cells. Vct cells showed an elevation in intracellular hypotaurine levels when exposed to 400 μM cysteamine during culturing. Western blot analysis was performed to assess the expression of the WNT5a protein in two cell lines treated with or without 400 μM cysteamine. The WNT5a protein was reduced specifically in Vct cells following exposure to cysteamine. **(D)** The Wnt5a mRNA expression level was verified by the RT-PCR analysis in 4 groups of cells.

movement and infiltration via the PI3K/Akt-RhoA pathway [21]. In gliomas, our findings suggest that Wnt5a expression is epigenetically regulated by hypotaurine, which may influence its functional role. Further studies are needed to delineate the specific signaling pathways involved and their impact on glioma progression.

In this assay, Wnt5a was identified as one of the most significantly perturbed genes in the extracellular matrix (ECM) receptor pathway (Fig 2C). Reduced levels of hypotaurine inside the cell promoted the activation of Wnt5a. In vitro studies showed that the use of cysteamine, a precursor of hypotaurine synthesis, led to higher levels of intracellular hypotaurine and improved cell invasion capabilities (Fig 3A and 3B). The observed decrease in intracellular Wnt5a proteins (Fig 3C) was accompanied by a similar decrease in the transcription level of Wnt5a mRNA (Fig 3D). The results supported the theory that Wnt5a functions as a cancer inhibitor in cells lacking the ability to synthesize hypotaurine.

Hypotaurine could inhibit demethylases. Epigenetic modification of Wnt5a promoter plays key roles in oncogenesis [12,14]. In this light, we hypothesized that promoter methylation might be one of the possible mechanism, through which hypotaurine attenuated Wnt5a's expression. In order to test the hypothesis, a demethylation agent 5-aza-2'-deoxycytidine was selected as control. 5-aza-2'-deoxycytidine can antagonize the methylation process as exhibited in Fig 4. As expected, the promoter methylation status was enhanced in hypotaurine treated ΔADO cells (Fig 4A). The specificity of hypotaurine's effects on demethylases remains an important question. Our findings suggest that hypotaurine inhibits demethylase activity, leading to increased methylation of the Wnt5a promoter. However, the precise mechanism by which hypotaurine exerts this effect is not fully understood. Hypotaurine may act directly on specific demethylases, such as TET enzymes, or it may influence broader epigenetic regulatory pathways. Previous studies have shown that hypotaurine can

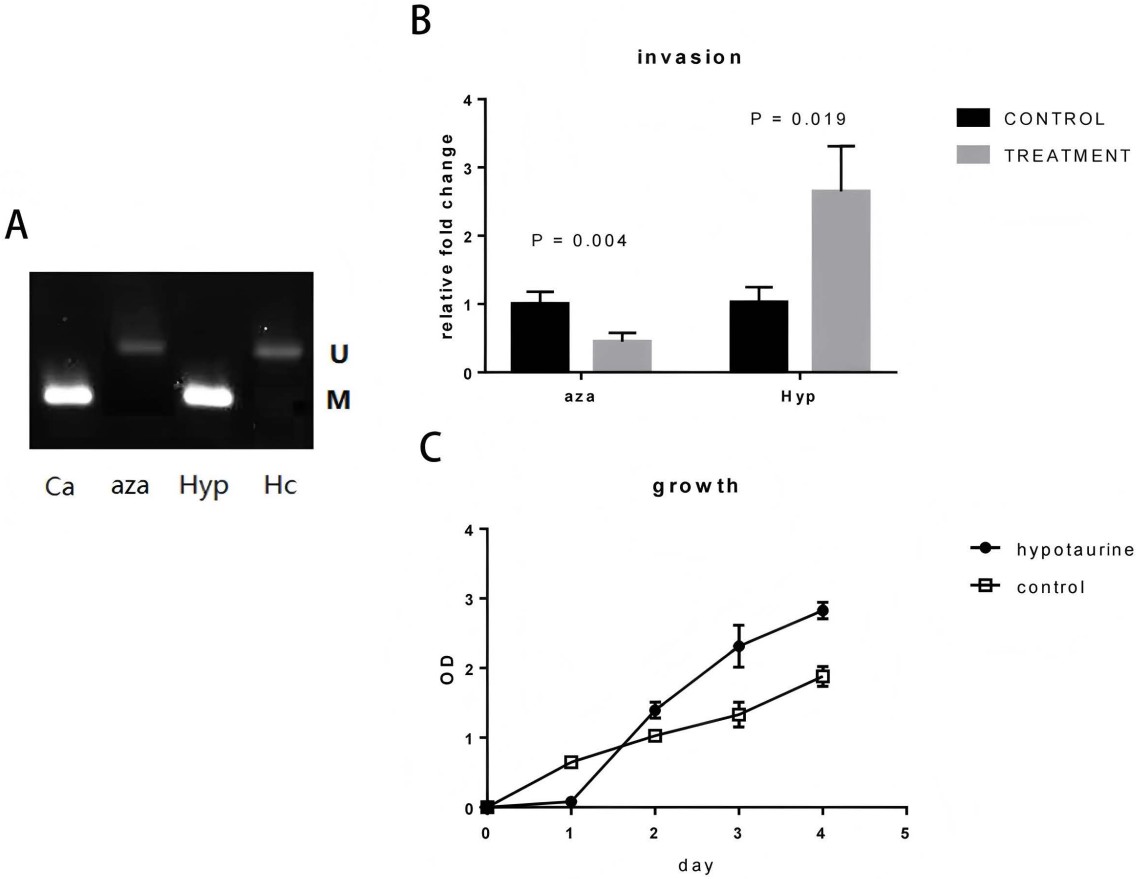

**Fig 4. Hypotaurine increased methylation status of Wnt5a promoter and cell invasion & proliferation abilities. (A)** Wnt5a promoter region methylation status evaluation of U251 cells by methylation-specific PCR analysis. The PCR was performed for each treatment using the two pairs of primers individually. For each treatment, the two PCR products were mixed together to be separated by gel electrophoresis. Ca: DMSO control; Aza: 5-aza-2'-deoxycytidine treated cells; Hyp: 10 μM hypotaurine treated; Hc: PBS treated control. U: unmethylated; M: methylated. **(B)** Differences in the invasion ability of ΔU251 cells treated with hypotaurine and 5-aza-2'-deoxycytidine. **(C)** Hypotaurine treatation promoted the proliferation abilitiy ofΔU251 cells.

modulate oxidative stress and redox balance, which are known to affect the activity of epigenetic enzymes [12]. It is also possible that hypotaurine's effects are not limited to Wnt5a but extend to other genes regulated by promoter methylation. Further studies using purified demethylase enzymes and genome-wide methylation analyses are needed to clarify the specificity and scope of hypotaurine's effects.

As previously described, the precise impact of Wnt5a on oncogenesis remained unclear. Many studies on gliomas tended to view Wnt5a as an invasion promoter [22,23]. The difference in findings between earlier studies and this research could be due to the use of a genetically altered cell line in this study. The involvement of the ADO gene in pathways beyond hypotaurine synthesis remained unconfirmed. To substantiate this conclusion, further investigation using authentic glioma samples is necessary.

The present findings are consistent with those of previous studies that have emphasised the role of epigenetic regulation in glioma invasion. For example, promoter methylation has been shown to regulate key genes involved in tumour progression, including genes in the Wnt signalling pathway. The observed effect of low taurine on Wnt5a promoter methylation suggests a novel mechanism by which glioma cells regulate their invasive behaviour. Targeting low taurine synthesis

or modulating Wnt5a promoter methylation may be a novel therapeutic strategy for the treatment of gliomas, and inhibitors of low taurine synthesis have the potential to reduce glioma cell invasion by altering the epigenetic regulation of Wnt5a. These findings provide a foundation for the development of new therapeutic strategies, such as the use of low-taurine inhibitors or epigenetic modulators. However, translating these findings into clinical practice is hindered by several challenges, including the need for selective and potent inhibitors, potential off-target effects, and the heterogeneity of glioma tumours. Future studies should focus on elucidating the exact molecular mechanisms underlying the effects of low taurine on methylation and invasion, including the identification of upstream regulators and downstream effectors. Moreover, further preclinical studies and in vivo models are required to evaluate the efficacy and safety of these approaches, thereby validating these findings and assessing their translational potential.

## Conclusions

This study demonstrates that hypotaurine promotes glioma cell invasion by epigenetically regulating Wnt5a expression through promoter methylation. These findings provide novel insights into the molecular mechanisms underlying glioma progression and suggest potential therapeutic strategies targeting hypotaurine synthesis or Wnt5a methylation. However, the study is limited by its reliance on in vitro models, and further in vivo studies are needed to validate these findings. Future research should focus on elucidating the signaling pathways involved, exploring the therapeutic potential of hypotaurine inhibitors, and addressing the challenges of translating these findings into clinical practice

## Supporting information

**S1 File. Raw images.**
(PDF)

## Author contributions

**Data curation:** Juan Liu.

**Methodology:** Qi Guan.

**Project administration:** Xiaoli Chu.

**Validation:** Ying Liu.

**Writing – original draft:** Hong Tian.

**Writing – review & editing:** Ying Liu.

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
