## [Decision Letter · Decision Letter 0]

25 Nov 2024

PONE-D-24-42277Hypotaurine promotes glioma cell invasion by hypermethylating the Wnt5a promoterPLOS ONE

Dear Dr. tian,

Thank you for submitting your manuscript to PLOS ONE. After careful consideration, we feel that it has merit but does not fully meet PLOS ONE’s publication criteria as it currently stands. Therefore, we invite you to submit a revised version of the manuscript that addresses the points raised during the review process.

We look forward to receiving your revised manuscript.

Kind regards,

Sayed Haidar Abbas Raza

Academic Editor

PLOS ONE

Journal requirements:    When submitting your revision, we need you to address these additional requirements. 1. Please ensure that your manuscript meets PLOS ONE's style requirements, including those for file naming. The PLOS ONE style templates can be found at https://journals.plos.org/plosone/s/file?id=wjVg/PLOSOne_formatting_sample_main_body.pdf and https://journals.plos.org/plosone/s/file?id=ba62/PLOSOne_formatting_sample_title_authors_affiliations.pdf 2. PLOS ONE now requires that authors provide the original uncropped and unadjusted images underlying all blot or gel results reported in a submission’s figures or Supporting Information files. This policy and the journal’s other requirements for blot/gel reporting and figure preparation are described in detail at https://journals.plos.org/plosone/s/figures#loc-blot-and-gel-reporting-requirements and https://journals.plos.org/plosone/s/figures#loc-preparing-figures-from-image-files. When you submit your revised manuscript, please ensure that your figures adhere fully to these guidelines and provide the original underlying images for all blot or gel data reported in your submission. See the following link for instructions on providing the original image data: https://journals.plos.org/plosone/s/figures#loc-original-images-for-blots-and-gels.   In your cover letter, please note whether your blot/gel image data are in Supporting Information or posted at a public data repository, provide the repository URL if relevant, and provide specific details as to which raw blot/gel images, if any, are not available. Email us at plosone@plos.org if you have any questions. 3. Please amend either the title on the online submission form (via Edit Submission) or the title in the manuscript so that they are identical. 4. Please match your authorship list in your manuscript file and in the system. 5. We noticed you have some minor occurrence of overlapping text with the following previous publication(s), which needs to be addressed: https://cdn.techscience.cn/uploads/attached/file/20191126/20191126061807_92356.pdf In your revision ensure you cite all your sources (including your own works), and quote or rephrase any duplicated text outside the methods section. Further consideration is dependent on these concerns being addressed. 6. We note that your Data Availability Statement is currently as follows: [All relevant data are within the manuscript and its Supporting Information files.] Please confirm at this time whether or not your submission contains all raw data required to replicate the results of your study. Authors must share the “minimal data set” for their submission. PLOS defines the minimal data set to consist of the data required to replicate all study findings reported in the article, as well as related metadata and methods (https://journals.plos.org/plosone/s/data-availability#loc-minimal-data-set-definition). For example, authors should submit the following data: - The values behind the means, standard deviations and other measures reported;- The values used to build graphs;- The points extracted from images for analysis. Authors do not need to submit their entire data set if only a portion of the data was used in the reported study. If your submission does not contain these data, please either upload them as Supporting Information files or deposit them to a stable, public repository and provide us with the relevant URLs, DOIs, or accession numbers. For a list of recommended repositories, please see https://journals.plos.org/plosone/s/recommended-repositories. If there are ethical or legal restrictions on sharing a de-identified data set, please explain them in detail (e.g., data contain potentially sensitive information, data are owned by a third-party organization, etc.) and who has imposed them (e.g., an ethics committee). Please also provide contact information for a data access committee, ethics committee, or other institutional body to which data requests may be sent. If data are owned by a third party, please indicate how others may request data access. 7. Please amend either the abstract on the online submission form (via Edit Submission) or the abstract in the manuscript so that they are identical.

Reviewers' comments:

Reviewer's Responses to Questions

**Comments to the Author**

1. Is the manuscript technically sound, and do the data support the conclusions?

Reviewer #1: Partly

Reviewer #2: Yes

2. Has the statistical analysis been performed appropriately and rigorously? 

Reviewer #1: Yes

Reviewer #2: Yes

3. Have the authors made all data underlying the findings in their manuscript fully available?

Reviewer #1: Yes

Reviewer #2: Yes

4. Is the manuscript presented in an intelligible fashion and written in standard English?

Reviewer #1: No

Reviewer #2: Yes

5. Review Comments to the Author

Reviewer #1: The paper presents a thorough exploration of glioma invasion mechanisms, focusing on the role of hypotaurine and its impact on Wnt5a gene expression, which adds valuable insights into glioma biology.The experimental design is strong, especially with the use of RNA sequencing to identify gene expression changes and how these changes correlate with cancer cell invasion. The connection between hypotaurine synthesis and its epigenetic effects on Wnt5a promoter methylation provides a novel perspective on glioma progression and offers potential therapeutic avenues. The use of specific inhibitors such as 5-aza-2’-deoxycytidine to explore the methylation status adds rigor and clarity to the findings.

However, several corrections need to be addressed, as outlined below:

1: Clearly define the research objective in the opening paragraph of the introduction. State the primary question being addressed (the role of hypotaurine in glioma cell invasion) upfront to provide a clear framework for the reader.

2: Provide clear definitions and background information on key terms early in the paper. For example, explain the biological significance of hypotaurine and Wnt5a and their roles in cancer biology. This will make the paper accessible to a broader audience.

3: Provide more mechanistic insights into how Wnt5a might function differently depending on cellular contexts. Include references to other studies where Wnt5a plays a role in cancer and provide a clearer explanation of its dual role as a tumor suppressor or promoter. This will clarify the ambiguity surrounding its function.

4: Expand the discussion on the potential clinical applications of these findings. Could targeting hypotaurine synthesis or Wnt5a methylation be a viable therapeutic strategy for glioma patients? Address the limitations and challenges in translating these findings from cell models to clinical practice.

5: Enhance the discussion by:

Interpreting the results in light of existing literature on glioma invasion and epigenetic regulation. How do these findings fit into the broader context of glioma biology?

Discussing the potential implications of these findings for therapeutic development (e.g., could hypotaurine inhibitors be used as a treatment strategy?).

Offering possible explanations for the observed effects of hypotaurine on methylation and invasion, and suggesting follow-up experiments to further elucidate these mechanisms.

6: Strengthen the conclusion by summarizing the key findings and emphasizing their potential impact on the understanding of glioma biology and the development of new therapeutic strategies. Additionally, clearly state any limitations of the current study and suggest areas for future research.

Reviewer #2: The manuscript by Tian et al studies the role of hypotaurine on the invasion and proliferation of glioma cells through its effect on Wnt5a expression. The authors used U251 glioma cells with reduced ADO expression. They observed that hypotaurine increases Wnt5a promoter methylation which results in reduced Wnt5a expression. This ultimately leads to enhanced tumor cell invasion and proliferation. The manuscript is well written and the conclusions are based on the results but I have a few concerns

1. The authors highlight the tumor suppressor role of Wnt5a but they should discuss the dual role of Wnt5a in more detail.

2. The authors should discuss the specificity of hypotaurine's effects on demethylases.

3. A graphical abstract will help the readers.

6. PLOS authors have the option to publish the peer review history of their article (what does this mean? ). If published, this will include your full peer review and any attached files.

**Do you want your identity to be public for this peer review?** For information about this choice, including consent withdrawal, please see our Privacy Policy .

Reviewer #1: **Yes: ** Simna Saraswathi Prasannakumari

Reviewer #2: No

---

## [Author Response · Author response to Decision Letter 0]

29 Mar 2025

Response to Reviewer #1 Comments

Reviewer Comment 1: Clearly define the research objective in the opening paragraph of the introduction. State the primary question being addressed (the role of hypotaurine in glioma cell invasion) upfront to provide a clear framework for the reader.

Response: Thank you for this suggestion. We have revised the opening paragraph of the introduction to clearly define the research objective and state the primary question being addressed. The revision can be seen in Lines 42, 45-46, page 2 in the revised manuscript.

Reviewer Comment 2: Provide clear definitions and background information on key terms early in the paper. For example, explain the biological significance of hypotaurine and Wnt5a and their roles in cancer biology. This will make the paper accessible to a broader audience.

Response: We have added definitions and background information on hypotaurine and Wnt5a in the introduction. The revision can be seen in Lines 67-74, page 2 and page 83-88, page3 in the revised manuscript.________________________________________

Reviewer Comment 3: Provide more mechanistic insights into how Wnt5a might function differently depending on cellular contexts. Include references to other studies where Wnt5a plays a role in cancer and provide a clearer explanation of its dual role as a tumor suppressor or promoter. This will clarify the ambiguity surrounding its function.

Response: We have expanded the discussion on Wnt5a’s dual role in cancer biology, citing relevant studies. The revision can be seen in Lines 255-271, page 9 in the revised manuscript.________________________________________

Reviewer Comment 4: Expand the discussion on the potential clinical applications of these findings. Could targeting hypotaurine synthesis or Wnt5a methylation be a viable therapeutic strategy for glioma patients? Address the limitations and challenges in translating these findings from cell models to clinical practice.

Response: We have expanded the discussion on the clinical implications of our findings. The revision can be seen in Lines 313-322, page 10 in the revised manuscript.________________________________________

Reviewer Comment 5: Enhance the discussion by:

• Interpreting the results in light of existing literature on glioma invasion and epigenetic regulation. How do these findings fit into the broader context of glioma biology?

• Discussing the potential implications of these findings for therapeutic development (e.g., could hypotaurine inhibitors be used as a treatment strategy?).

• Offering possible explanations for the observed effects of hypotaurine on methylation and invasion, and suggesting follow-up experiments to further elucidate these mechanisms.

Response: We have enhanced the discussion by integrating the results with existing literature, interpreting the findings, and suggesting follow-up experiments. The revision can be seen in Lines 305-322, page 10 in the revised manuscript.________________________________________

Reviewer Comment 6: Strengthen the conclusion by summarizing the key findings and emphasizing their potential impact on the understanding of glioma biology and the development of new therapeutic strategies. Additionally, clearly state any limitations of the current study and suggest areas for future research.

Response: We have revised the conclusion to summarize the key findings, highlight their significance, and address limitations and future directions. The revision can be seen in Lines 324-331, page 10 in the revised manuscript.

Response to Reviewer #2 Comments

Reviewer Comment 1: The authors highlight the tumor suppressor role of Wnt5a but they should discuss the dual role of Wnt5a in more detail.

Response: Thank you for pointing this out. We have expanded the discussion section to provide a more detailed explanation of Wnt5a's dual role in cancer biology. The revision can be seen in Lines 247-264, page 9 in the revised manuscript.________________________________________

Reviewer Comment 2: The authors should discuss the specificity of hypotaurine's effects on demethylases.

Response: We appreciate this suggestion and have added a discussion on the specificity of hypotaurine's effects on demethylases. The revision can be seen in Lines 287-297, page 9-10 in the revised manuscript.________________________________________

Reviewer Comment 3: A graphical abstract will help the readers.

Response: Thank you for this suggestion. We have created a graphical abstract summarizing the key findings of the study. The graphical abstract visually illustrates the role of hypotaurine in promoting glioma cell invasion and proliferation through its effects on Wnt5a promoter methylation and expression. The graphical abstract has been included as a separate figure in the manuscript.

---

## [Decision Letter · Decision Letter 1]

10 Apr 2025

Hypotaurine promotes glioma cell invasion by hypermethylating the Wnt5a promoter

PONE-D-24-42277R1

Dear Dr. tian,

We’re pleased to inform you that your manuscript has been judged scientifically suitable for publication and will be formally accepted for publication once it meets all outstanding technical requirements.

Kind regards,

Sayed Haidar Abbas Raza

Academic Editor

PLOS ONE

Additional Editor Comments (optional):

Reviewers' comments:

Reviewer's Responses to Questions

**Comments to the Author**

1. If the authors have adequately addressed your comments raised in a previous round of review and you feel that this manuscript is now acceptable for publication, you may indicate that here to bypass the “Comments to the Author” section, enter your conflict of interest statement in the “Confidential to Editor” section, and submit your "Accept" recommendation.

Reviewer #1: All comments have been addressed

Reviewer #2: All comments have been addressed

2. Is the manuscript technically sound, and do the data support the conclusions?

Reviewer #1: Yes

Reviewer #2: Yes

3. Has the statistical analysis been performed appropriately and rigorously? 

Reviewer #1: Yes

Reviewer #2: Yes

4. Have the authors made all data underlying the findings in their manuscript fully available?

Reviewer #1: Yes

Reviewer #2: Yes

5. Is the manuscript presented in an intelligible fashion and written in standard English?

Reviewer #1: Yes

Reviewer #2: Yes

6. Review Comments to the Author

Reviewer #1: (No Response)

Reviewer #2: The authors have answered all the comments point by point in a satisfactory manner. They have also included the changes in the manuscript.

7. PLOS authors have the option to publish the peer review history of their article (what does this mean? ). If published, this will include your full peer review and any attached files.

**Do you want your identity to be public for this peer review?** For information about this choice, including consent withdrawal, please see our Privacy Policy .

Reviewer #1: No

Reviewer #2: No

---

## [Editor Report · Acceptance letter]

PONE-D-24-42277R1

PLOS ONE

Dear Dr. Tian,

I'm pleased to inform you that your manuscript has been deemed suitable for publication in PLOS ONE. Congratulations! Your manuscript is now being handed over to our production team.

Kind regards,

on behalf of

Dr. Sayed Haidar Abbas Raza

Academic Editor

PLOS ONE